# Transitions in metabolic syndrome and metabolic obesity status over time and risk of urologic cancer: A prospective cohort study

Xia Wang[1], Runxue Jiang[2,3]*, Jianglun Shen[2], Shuohua Chen[4], Shouling Wu[4], Hailong Hu[3], Haifeng Cai[2]

**1** Department of Gynaecology, Tangshan Hongci Hospital, Tangshan, Hebei, China, **2** Department of Oncology Surgery, Tangshan People's Hospital, Tangshan, Hebei, China, **3** Department of Urology, Tianjin Institute of Urology, The Second Hospital of Tianjin Medical University, Tianjin, China, **4** Health Department of Kailuan(Group), Tangshan, Hebei, China

☯ These authors contributed equally to this work.
* cnhbjirx@163.com

**Data Availability Statement:** The data supporting the findings of this study contain potentially identifying participant information. Due to ethical and legal restrictions, these data cannot be made

## Abstract

### Background and aims

The effects of metabolic obesity (MO) phenotypes status and their dynamic changes on urologic cancer (UC) is ignored. We aimed to investigate the association between metabolic syndrome (MetS) and MO status at baseline, their dynamic changes and UC risk.

### Methods

This paper studied 97,897 subjects who were free of cancers at baseline (2006−2007). Individuals were classified into four MO phenotypes by MetS and obesity at baseline. Transitions in MetS and MO status from 2006–2007 to 2008–2009 were considered. The hazard ratios (HRs) and 95% confidence intervals (CIs) for UC were assessed by multifactorial Cox proportional risk regression models. The main limitations of this study are as follows: the ratio of men to women in the cohort is unbalanced; the impacts of MetS and MO on each cancer type (kidney cancer, prostate cancer, bladder cancer) have not been analyzed separately; the transition intervals of MetS and MO phenotypes are relatively short.

### Results

From baseline (2006–2007) survey to December 31, 2020, during a median follow-up of 14.02 years, 554 cases of UC were diagnosed. Participants with MetS [HRs (95% CI) = 1.26 (1.06–1.49)] and metabolically unhealthy obesity (MUO) [HRs (95% CI) = 1.49 (1.17–1.89)] had significantly higher risk of UC than those with non-MetS and metabolically healthy normal weight (MHN). Transitions in MetS and MO phenotypes over time were studied. Compared with non-MetS to non-MetS, the risks for UC in MetS to MetS [HRs (95% CI) = 1.45 (1.11–1.88)] was increased. Compared with MHN to MHN, both MUO to metabolically healthy obesity (MHO) [HRs (95% CI) = 2.65 (1.43–4.92)] and MUO to MUO [HRs (95% CI) = 1.60 (1.06–2.42)] had significantly higher UC risk.

publicly available. However, researchers who meet the criteria for accessing confidential data can obtain the data through the Kailuan Study at kailuancsh@163.com.

**Funding:** The author(s) received no specific funding for this work.

**Competing interests:** The authors have declared that no competing interests exist.

## Conclusions

MetS and MUO increased the UC risk at baseline. Transitions of MetS to MetS, MUO to MUO and even MUO to MHO over time significantly increased the risk of UC development.

## Introduction

Worldwide, urologic malignancies pose a serious threat to human health. Based on GLOBO-CAN data, in 2020 alone, the number of new cases of prostate, bladder and kidney cancers reached 1,414,259, 573,278 and 431,288, respectively. In the same year, 375,304, 212,536 and 179,368 deaths resulted from prostate, bladder and kidney cancers, respectively [1]. Although urologic cancer (UC) is highly dangerous, its etiology remains unclear, and known risk factors include age, race, family history of malignancy and smoking [2–4].

Metabolic syndrome (MetS) is a comprehensive syndrome involving blood pressure, glucose and lipid abnormalities, and central obesity. A meta-analysis showed MetS is associated with an increased risk of colorectal, endometrial and postmenopausal breast cancer as well as overall cancers in humans [5] but not with gastric [6] and lung [7] cancer. Mili et al. considered that the association between MetS and cancer may vary depending on the site of cancer [8]. What is the effect of metabolic health status on urinary system cancer? Some studies have found that MetS is associated with an increased risk of bladder, prostate and kidney cancers [9–11]. However, some reports are conflicting [12, 13].

Obesity is a well-established risk factor for kidney [14–16], bladder [17, 18] and prostate [19] cancers. Obesity can result from the adverse effects of metabolic abnormalities, such as hyperglycaemia, hypertension and hypercholesterolaemia [20]. However, not all obese individuals have the characteristics of metabolic disorders. Body mass index (BMI) is a common criterion for identifying obesity. These conditions of individuals with or without MetS and obesity are classified into several metabolic obesity (MO) phenotypes, including metabolically unhealthy obesity (MUO), metabolically healthy obesity (MHO), and so on [21, 22]. Studies have shown that different MO phenotypes exhibit different risks of cancer onset, a increased cancer risk for MUO compared to MHO regardless of population heterogeneity, or the definitions of obesity and metabolic status [23]. Existing studies on the association between the combination of metabolic health status and obesity and UC are scarce. Do different MO phenotypes have different effects on the risk of UC?. So, to prevent development of the whole UC, to screen out high-risk groups of UC, investigating the effects of MetS, MO phenotypes, and changes in MetS and MO phenotypes over time on UC is necessary. Therefore, in the present paper, a cohort study based on the Kailuan Study is used to investigate the association between the transitions in MetS, MO phenotypes over time and the risk of UC to provide a theoretical basis for the identification of risk factors and the scientific prevention and control of UC.

## Population and methods

### Ethics approval and consent to participate

The study was conducted in accordance with the Declaration of Helsinki and was approved by the Ethics Committee of Kailuan General Hospital, Ethics number KS-2006-5. The participants provided their written informed consent to participate in this study.

## Study cohort

The Kailuan Study is an ongoing cohort study based on functional community population that began in July 1, 2006. All the participants underwent questionnaires interviews, clinical examinations and laboratory tests every 2 years at 11 Kailuan Group affiliated hospitals by well-trained physicians or nurses using a standardized protocol, and follow-up data were obtained for events including UC, allowing us to investigate the relationship between transitions in MetS and MO phenotypes and UC. More details about the Kailuan Study can be found elsewhere [24, 25]. Participants who met the following criteria were recruited in this study: (1) included in the baseline survey (2006–2007) population of the Kailuan Study, (2) age ≥18 years and (3) signed the written informed consent form. The exclusion criterion was history of malignancy.

A total of 101,510 participants (81,110 males and 20,400 females; aged 18–98 years) met the criteria and were recruited in this study, and 379 participants with a history of cancer at baseline (2006–2007), 1,468 participants with missing data on BMI and 1,766 participants with missing data on MetS status were excluded. Finally, 97,897 participants were included in the analysis of the association between MetS and MO status at baseline and the risk of UC. To explore the effects of transitions in MetS and MO status over time (from 2006–2007 to 2008–2009) on the risk of UC, 35,178 participants, amongst whom 24,162 did not participate in the 2008–2009 survey, 34 with incident UC between the 2006–2007 and 2008–2009 surveys, 3,186 participants with missing data on BMI and 7,796 participants with missing data on MetS components in the 2008–2009 survey were further excluded, leaving 62,719 participants for assessing the association of transitions in MetS and MO status and risk of UC (Fig 1).

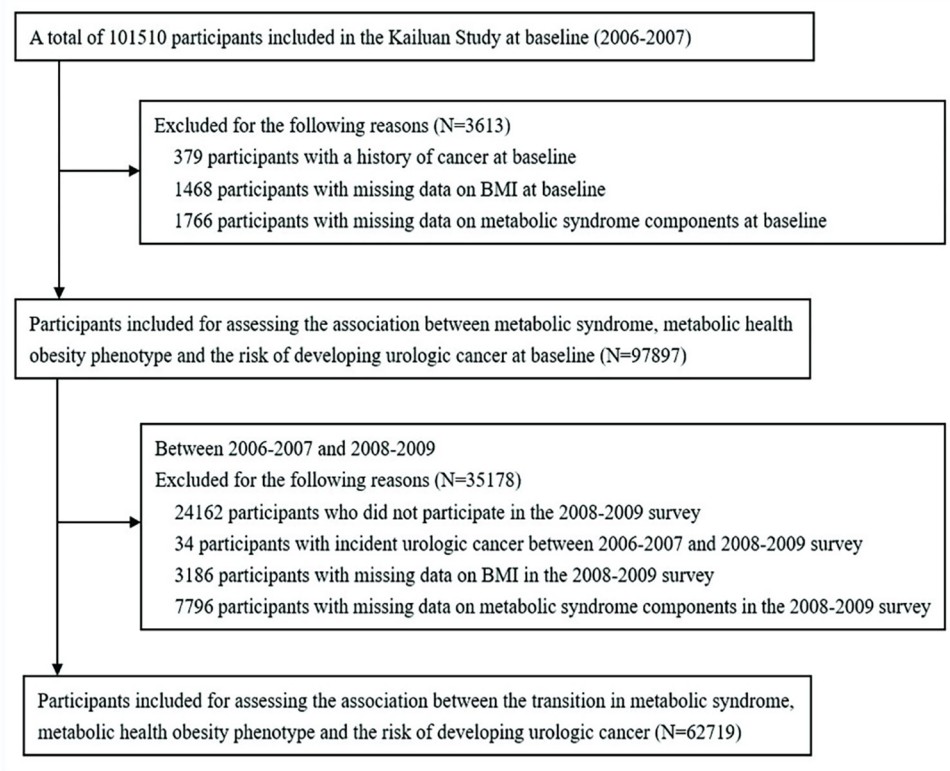

**Fig 1. Flow diagram of the study population.**

## Collection of exposure information

The baseline data included sociodemographic characteristics (age, gender, occupation, education level, economic income and marital status), lifestyle characteristics (smoking status, alcohol consumption, salt intake and sitting time), history of previous diseases, physical examination data [body weight, height, waist circumference (WC) and blood pressure (BP)] and blood indices (fasting glucose and lipids).

Participants wore light clothing, removed their shoes and hats, and had their body weight and height measured. BMI was calculated by weight in kilograms divided by height in square meters. WC was measured at the level of the midpoint between the anterior superior iliac crest and the lower rib cage. Systolic and diastolic blood pressure were measured using a mercury sphygmomanometer with a suitable cuff on the left arm of the subject after 5 min of rest and then again after 5 min, and the average of the two measurements was recorded. Early morning fasting blood samples were collected from the subjects to measure blood glucose and lipids. Fasting blood glucose level was measured using the hexokinase/glucose-6-phosphate dehydrogenase method, and the coefficient of variance of blind quality control samples was < 2.0%. Triglycerides were determined by glycerol phosphate oxidase assay (coefficient of mutual variation < 10%). After the precipitation of apolipoprotein B with dextrose sulphate and magnesium chloride, high-density lipoprotein cholesterol level was measured in the supernatant [25].

## Definition of variables

MetS was defined according to the harmonized International Diabetes Federation criteria [26], as shown in Table 1. Obesity statuses were categorized according to the Working Group on Obesity in China into normal weight (BMI < 28 kg/m$^2$) and obesity (BMI ≥ 28 kg/m$^2$) [27]. MO status was classified into four phenotypes according to whether the subject had MetS or obesity (Table 1).

According to MetS and obesity status at baseline (2006–2007) and 2008–2009 survey, transitions in MetS were divided into four phenotypes: non-MetS to non-MetS, non-MetS to

**Table 1. Definitions of MetS and MO phenotypes.**

| Harmonized International Diabetes Federation criteria for MetS: ≥3 of the following components: |
|---|
| (1) central obesity, defined as WC ≥ 90 cm in men and WC ≥ 80 cm in women |
| (2) elevated TG, defined as TG ≥ 150 mg/dL (1.7 mmol/L) and/or drug treatment for elevated TG |
| (3) reduced HDL-C, defined as HDL-C < 40 mg/dL (1.0 mmol/L) in males and HDL-C < 50 mg/dL (1.3 mmol/L) in females and/or durg treatment for reduced HDL-C |
| (4) elevated BP, defined as systolic BP ≥ 130 mmHg and/or diastolic BP ≥ 85 mmHg, and/or a history of hypertension treated with antihypertensive drugs |
| (5) elevated FBG, defined as FBG ≥ 100 mg/dL (5.6 mmol/L) and/or receiving glucose-lowering medication for elevated glucose |
| Metabolic obesity phenotypes: |
| MHN: BMI<28 without MetS |
| MUN: BMI<28 with MetS |
| MHO: BMI≥28 without MetS |
| MUO: BMI≥28 with MetS |

Abbreviations: MetS, metabolic syndrome; MO, metabolic obesity; WC, waist circumference; TG, triglycerides; HDL-C, high-density lipoprotein cholesterol; BP, blood pressure; FBG, fasting blood glucose; MHN, metabolically healthy normal weight; MUN, metabolically unhealthy normal weight; MHO, metabolically healthy obesity; MUO, metabolically unhealthy obesity; BMI, body mass index.

MetS, MetS to non-MetS and MetS to MetS. Transitions in MO status were classified into seven phenotypes: MHN to MHN, MHO to MHO, MHO to MUO, MUN to MUO, MUO to MHO, MUO to MUN and MUO to MUO.

Smoking status was classified as follows: never smoker, former smoker (has quit smoking for more than 12 months) and current smoker (smokes one or more cigarettes per week for not less than 12 consecutive months). Alcohol consumption status was divided into the following categories: never, former (abstained from alcohol for more than six months) and current alcohol consumption (one or more drinks per month for no less than six months in a row).

## Collection of endpoint event information

The follow-up period started when the participants were recruited and the baseline survey was performed (2006–2007). The last follow up was completed on December 31, 2020. The follow-up endpoint event was a new UC or death in the observed subject (whichever came first). Firstly, the information of participants' medical records was obtained from the Tangshan City Health Insurance System. Professionally trained investigators then went to the hospitals to collect information on the subjects' medical history. Clinicians verified the pathology, imaging (including magnetic resonance imaging, computed tomography and colour Doppler ultrasonography) and blood biochemical examination results to confirm and refine the diagnosis of UC. Tumour cases were encoded according to the International Classification of Diseases-10. UC included prostate cancer, kidney cancer, carcinoma renal pelvis, ureteral cancer, bladder cancer and urethral cancer with codes C61 and C64–C68. Information on death events was obtained from the Kailuan Group Social Insurance System.

## Statistical analysis

The baseline characteristics of MetS, MO phenotypes, and transitions in MetS and MO status were determined. Continuous variables were expressed as mean ± standard deviation (SD), whereas categorical variables were expressed as the number of cases (percentage). Multiple imputation methods were used to replace the missing data for covariates.

Multifactorial Cox proportional risk regression models were used to analyse the relationships between MetS (non-MetS group as the reference) and MO phenotypes (MHN group as the reference) at baseline (2006–2007) and the risk of developing UC. The association between the transitions in MetS (compared with non-MetS to non-MetS individuals) and MO status (compared with MHN to MHN individuals) from baseline (2006–2007) to 2008–2009 survey and the risk of UC incidence during the follow up (after 2008–2009) was also assessed in the same way. Model 1 was adjusted for age (at the start of follow up) and gender, whereas Model 2 was further adjusted for smoking status, alcohol consumption, occupation, education level, income, marital status, salt intake and sitting time.

Subgroup analyses were conducted to explore the associations of MetS, MO status and their transitions with the risk of UC according to age, gender and smoking status, Multiplication interactions were performed as well.

Several sensitivity analyses were performed to examine the robustness of the results excluding participants who (1) had a history of myocardial infarction and stroke, (2) developed UC within the first 2 years of follow up and (3) had less than 1.5 years between baseline (2006–2007) and 2008–2009 surveys (only for transitions in MetS and MO status).

All statistical analyses were performed using SAS 9.4 and were considered statistically significant at $P < 0.05$ (two sided).

## Results

### Baseline characteristics

The baseline characteristics of the participants classified by MetS and MO status at baseline (2006–2007) and transition in MetS and MO status (2006–2007 to 2008–2009) are shown in Table 2 and S1–S3 Tables. In the present cohort, at the baseline (2006–2007) survey, the mean age of the participants was 51.76 ± 12.56 years, and 78,285 (79.97%) were male. A total of 31,336 (32.01%) and 11,046 (11.28%) subjects were defined as MetS and MUO, respectively. Compared with the non-MetS and MHN, the mean age and the proportions of males, subjects

**Table 2. Baseline characteristics of participants by MetS status, 2006–2007.**

| Characteristics | Total cohort | Non-MetS | MetS | p |
|---|---|---|---|---|
|  | (n = 97897) | (n = 66561) | (n = 31336) |  |
| Age(years,mean±SD) | 51.76±12.56 | 50.46±12.86 | 54.51±11.41 | <0.0001 |
| Gender, n(%) |  |  |  |  |
| Female | 19612(20.03) | 13969(20.99) | 5643(18.01) | <0.0001 |
| Male | 78285(79.97) | 52592(79.01) | 25693(81.99) |  |
| Smoking status, n(%) |  |  |  |  |
| Never | 58327(59.58) | 40165(60.34) | 18162(57.96) | <0.0001 |
| Former | 5867(5.99) | 3463(5.20) | 2404(7.67) |  |
| Current | 33703(34.43) | 22933(34.45) | 10770(34.37) |  |
| Alcohol consumption, n(%) |  |  |  |  |
| Never | 57442(58.68) | 39332(59.09) | 18110(57.79) | <0.0001 |
| Former | 4003(4.09) | 2362(3.55) | 1641(5.24) |  |
| Current | 36452(37.24) | 24867(37.36) | 11585(36.97) |  |
| Occupation, n(%) |  |  |  |  |
| White collar | 7884(8.05) | 5269(7.92) | 2615(8.35) | 0.0214 |
| Blue collar | 90013(91.95) | 61292(92.08) | 28721(91.65) |  |
| Education level, n(%) |  |  |  |  |
| Illiteracy and primary | 10964(11.20) | 6707(10.08) | 4357(13.59) | <0.0001 |
| Middle school | 79901(81.62) | 54560(81.97) | 25341(80.87) |  |
| College and above | 7032(7.18) | 5294(7.95) | 1738(5.55) |  |
| Income(yuan per person per month), n(%) |  |  |  |  |
| <600 | 27897(28.50) | 18575(27.91) | 9322(29.75) | <0.0001 |
| ≥600-<1000 | 62410(63.75) | 42881(64.42) | 19529(62.32) |  |
| ≥1000 | 7590(7.75) | 5105(7.67) | 2485(7.93) |  |
| Marital status, n(%) |  |  |  |  |
| Single | 4420(4.51) | 3197(4.80) | 1223(3.90) | <0.0001 |
| Married/cohabiting | 93477(95.49) | 63364(95.20) | 30113(96.10) |  |
| Salt intake, n(%) |  |  |  |  |
| Light | 9201(9.40) | 6240(9.37) | 2961(9.45) | <0.0001 |
| General | 78122(79.80) | 53635(80.58) | 24487(78.14) |  |
| Heavy | 10574(10.80) | 6686(10.04) | 3888(12.41) |  |
| Sitting time(h/day), n(%) |  |  |  |  |
| <4 | 72714(74.28) | 49721(74.70) | 22993(73.38) | <0.0001 |
| ≥4-<8 | 22070(22.54) | 14746(22.15) | 7324(23.37) |  |
| ≥8 | 3113(3.18) | 2094(3.15) | 1019(3.25) |  |

Abbreviations: MetS, metabolic syndrome.

**Table 3. Associations between MetS and MO status at baseline (2006–2007) and risk of UC.**

| Groups | | Total | Person | Incident | HR(95%CI) | |
|---|---|---|---|---|---|---|
| | | cases | years | cases | Model 1 | Model 2 |
| MetS status | Non-MetS | 66561 | 886385.16 | 332 | Ref | Ref |
| | MetS | 31336 | 406297.93 | 222 | 1.25(1.05–1.48) | 1.26(1.06–1.49) |
| MO status | MHN | 59385 | 790282.21 | 303 | Ref | Ref |
| | MHO | 7176 | 96102.96 | 29 | 0.87(0.60–1.28) | 0.89(0.61–1.30) |
| | MUN | 20290 | 261965.90 | 136 | 1.12(0.92–1.37) | 1.130(0.92–1.38) |
| | MUO | 11046 | 144332.02 | 86 | 1.49(1.15–1.86) | 1.49(1.17–1.89) |

Abbreviations: MetS, metabolic syndrome; MO, metabolic obesity; UC, urologic cancer; MHN, metabolically healthy normal weight; MHO, metabolically healthy obesity; MUN, metabolically unhealthy normal weight; MUO, metabolically unhealthy obesity; HR, hazard ratio; CI, confidence interval; Ref, reference.

Model 1 was adjusted for age and gender; Model 2 was further adjusted for smoking status, alcohol consumption, occupation, education level, income, marital status, salt intake and sitting time.

with low education level (illiteracy and primary) and heavy salt intake were higher, whereas the proportions of nonsmokers, nondrinkers and subjects with low sitting time (< 4 h/day) were lower in MetS and MUO. MetS to MetS and MUO to MUO compared with non-MetS to non-MetS and MHN to MHN showed the demographic and socioeconomic characteristics of individuals who participated at baseline (2006–2007), and the 2008–2009 survey was consistent with the above.

## Association between MetS, MO status at baseline and risk of UC

In this study, the cumulative follow-up period was 1,292,683.09 person–years, the median follow-up time was 14.02 years and 554 participants developed UC from the baseline. Cox proportional hazards regression models were used to assess the effect of MetS and MO status on the development of UC. Individuals with MetS or MUO had significantly higher risk of UC than non-MetS and MHN, and the hazard ratios (HRs) and 95% confidence intervals (95% CI) were 1.26 (1.06–1.49) and 1.49 (1.17–1.89), respectively (Table 3). Additionally, in analyses stratified by age, gender and smoking status, the effect of MetS, MO status at baseline on the risk of UC by age displayed a significant difference (p for interaction < 0.05). Subgroup analyses indicated low-age (< 55 years old), male subjects, and former and current smokers with MetS and MUO at baseline had an increased risk of UC than non-MetS and MHN (S4 and S5 Tables).

## Association between transition in MetS, MO status and risk of UC

During a median follow-up time of 11.96 years, from the 2008–2009 survey to December 31, 2020, 316 UC events occurred. For the effect of transition in MetS on the development of UC, compared with non-MetS to non-MetS, MetS to MetS [HRs (95% CI) = 1.45 (1.11–1.88)] had a higher risk of UC, whereas MetS to non-MetS [HRs (95% CI) = 1.29 (0.92–1.82)] had a no statistically significant difference; for transition in MO status, compared with MHN to MHN, MUO to MUO [HRs (95% CI) = 1.60 (1.06–2.42)] had a significantly higher risk of UC, and similarly, MUO to MHO [HRs (95% CI) = 2.65 (1.43–4.92)] showed a higher risk of UC (Table 4). Moreover, in analyses stratified by age, gender and smoking status, the effect of transition in MetS on the risk of UC by age displayed a significant difference (p for interaction < 0.05). Similarly, for transitions in MetS and MO status, the risk of developing UC of subjects with MetS to MetS and MUO to MUO were significantly higher than non-MetS to non-MetS and MHN to MHN in low-age (< 55 years old), male, and former and

**Table 4. Associations between transitions in MetS and MO status (2006–2007 to 2008–2009) and risk of UC.**

| Groups | | Total | Person | Incident | HR(95%CI) | |
|---|---|---|---|---|---|---|
| | | cases | years | cases | Model 1 | Model 2 |
| Transition in MetS status | | | | | | |
| MetS status at baseline (2006–2007) | MetS status at follow-up (2008–2009) | | | | | |
| Non-MetS | Non-MetS | 32826 | 378952.69 | 139 | Ref | Ref |
| Non-MetS | MetS | 9905 | 112711.12 | 41 | 0.84(0.60–1.20) | 0.85(0.60–1.21) |
| MetS | Non-MetS | 7025 | 78977.25 | 44 | 1.28(0.91–1.79) | 1.29(0.92–1.82) |
| MetS | MetS | 12963 | 144280.66 | 92 | 1.42(1.09–1.85) | 1.45(1.11–1.88) |
| Transition in MHO status | | | | | | |
| MHO status at baseline (2006–2007) | MHO status at follow-up (2008–2009) | | | | | |
| MHN | MHN | 29218 | 337247.96 | 121 | Ref | Ref |
| MHO | MHO | 1437 | 16723.51 | 4 | 0.80(0.30–2.17) | 0.82(0.30–2.22) |
| MHO | MUO | 1423 | 16445.79 | 8 | 1.34(0.65–2.74) | 1.34(0.65–2.74) |
| MUN | MUO | 958 | 10503.11 | 7 | 1.40(0.66–3.01) | 1.42(0.66–3.05) |
| MUO | MHO | 1005 | 11443.27 | 11 | 2.60(1.40–4.82) | 2.65(1.43–4.92) |
| MUO | MUN | 1355 | 15154.18 | 10 | 1.55(0.81–2.95) | 1.56(0.82–2.97) |
| MUO | MUO | 3992 | 44897.53 | 28 | 1.57(1.04–2.37) | 1.60(1.06–2.42) |

Abbreviations: MetS, metabolic syndrome; MO, metabolic obesity; UC, urologic cancer; MHN, metabolically healthy normal weight; MHO, metabolically healthy obesity;MUN, metabolically unhealthy normal weight; MUO, metabolically unhealthy obesity; HR, hazard ratio; CI, confidence interval; Ref, reference.

Model 1 was adjusted for age and gender; Model 2 was further adjusted for smoking status, alcohol consumption, occupation, education level, income, marital status, salt intake and sitting time.

current smoking subgroups, whereas in former and current smokers, participants with MetS to non-MetS, the effect still had statistically significant differences (S6 and S7 Tables).

### Sensitivity analyses of the relationship between MetS, MO status, and their transitions and the risk of UC

In the sensitivity analyses, the associations of MetS, MO status, and their transitions and the risk of developing UC did not materially change compared with the main models after excluding participants with a history of myocardial infarction and stroke, subjects who developed UC within the first two years of follow up or individuals with less than 1.5 years between baseline (2006–2007) and 2008–2009 surveys (only for the transitions in MetS and MHO status), as shown in S8 and S9 Tables.

## Discussion

Amongst Kailuan Study participants, having MetS and MUO at baseline was associated with a risk for incident UC; the transitions in MetS and MO status over time (2006–2007 to 2008–2009) revealed the risk of UC was strongly higher in subjects with MetS to MetS, MUO to MHO and MUO to MUO. Individuals with MetS to non-MetS had a higher risk, but the difference was not statistically significant.

The association between MetS and MO status and the risk of UC has not been reported in previous studies. In this paper, MetS at baseline increased the risk of developing UC by 26% compared with the non-MetS population. This result was similar to that observed in studies of a certain part of the UC, such as bladder [9], prostate [10] and kidney [11] cancer, although some inconsistent research reports [12, 13] were noted. These differences in findings may be explained by dissimilarities in the observed populations, races, study designs, the length of the

follow-up period, and adjustments for confounding factors and definitions of MetS. MetS and obesity frequently coexist, and obesity is a risk factor for UC [14–19]. Amongst MO phenotypes, MUO subjects had a 49% increased risk of UC at baseline than MHN subjects. Studies by Kim JW et al. also showed that participants with metabolically unhealthy obesity had a higher risk of prostate cancer [28] and bladder cancer [29]. MetS and MO status can be changed over time and with certain interventions. In a subsequent study, the influence of transitions in MetS and MO status (2006–2007 to 2008–2009) on the risk of developing UC was explored. It showed that compared with non-MetS to non-MetS, the UC risk of MetS to MetS significantly increased by 45%. The risk of developing UC of participants with MetS at baseline (2006–2007) that changed into non-MetS (2008–2009 survey) was higher, but the difference was not statistically significant. These results confirm MetS is a risk factor of UC, and the improvement of MetS can reduce the risk of UC, which provides a scientific basis for the prevention of UC. In the study of MO status transition, compared with MHN to MHN, the UC risk of MUO to MUO was significantly increased [HR (95% CI) = 1.60 (1.06–2.42)]. In a study in South Korea, it is also shown that individuals who continuously stay in a state of metabolically unhealthy obesity have a greater risk of developing incident kidney cancer compared to those in the stable metabolically healthy non-obese [30]. Moreover, when the status of MUO changed to MHO, the metabolic unhealth status improved, but the UC risk was still significantly higher. Therefore, the reduction of BMI in patients with MUO whilst controlling MetS is of realistic significance to prevent the occurrence of UC.Subgroup analyses showed that in low-age (< 55 years old), male, and former and current smoking subgroup, baseline MetS and MUO and transitions of MetS to MetS and MUO to MUO significantly increased the risk of UC compared with controls, especially in the smoking subgroup, and even the effect of MetS to non-MetS on the incident of UC was statistically significant. Such results suggest that the prevention and control of MetS and obesity should be strengthened in low-age, male individuals. Moreover, the harm of tobacco should be reduced, and smoking bans are of great significance for the prevention and control of UC.

The results of MetS and MUO on promoting the development of cancer can be explained by several plausible pathophysiological mechanisms: Firstly, they possess numerous shared risk elements, such as advanced age, obesity, lack of physical activity, an unhealthful diet, disruption of the biological clock, oxidative stress, air pollution, and damage resulting from exposure to substances that interfere with the endocrine system [31, 32]. Secondly, MetS and obesity are frequently marked by insulin resistance and hyperinsulinemia [33–35]. Insulin resistance prompts the generation of reactive oxygen species that are capable of damaging DNA and facilitating malignant transformation [36]. Hyperinsulinemia augments the biological activity of IGF-1 [37]. IGF-1 induces and activates the Ras/Raf/MAPK and PI3K/Akt/mTOR pathways, thereby reducing apoptosis, promoting cell proliferation and survival, and elevating the risk of tumor development [38]. Thirdly, obesity may give rise to the infiltration of immune cells like macrophages and lymphocytes. These cells serve as crucial sources of circulating pro-inflammatory factors (tumor necrosis factor-α and interleukin-6). High concentrations of pro-inflammatory mediators drive cancer development and progression via local and systemic impacts [39–41]. The state of obesity is marked by a high ratio of leptin to adiponectin. Adiponectin has an inhibitory impact on cell proliferation and metastasis, while leptin stimulates cell proliferation and promotes invasion and migration [42]. Moreover, increased angiogenic factors in patients with hypertension might be related to the risk of renal malignancies developing [43]. Additionally, chronic hyperglycemia can also lead to oxidative damage to cellular DNA [44], which can result in worse tumor grading, higher metastatic potential, and resistance to chemotherapy [45].

## Strengths and limitations

The strengths of this study are large sample size, prospective cohort design, robust follow-up mechanism, long follow-up period, detailed analysis of MetS and MO status transitions, and comprehensive statistical adjustments for potential confounders. Our study also has some limitations. Firstly, the female participants accounted for a lower percentage (20.03%) than the male participants, and the gender imbalance may influence the generalizability of the results. Secondly, although the effects of transition of MetS and MO phenotypes on overall UC have been explored, while we failed to analyze each cancer type (kidney cancer, prostate cancer, bladder cancer) separately because of the limited number of cases. Thirdly, the transition intervals of MetS and MO phenotypes were relatively short. The sensitivity analysis excluded participants whose interval between two surveys was less than 1.5 years. Although the results were still stable, if the follow up continues and the transition intervals are lengthened, it will be more meaningful for explaining the results. Solutions to limitations and future research directions are to establish a dynamic cohort to expand the number of observation objects and prolong the follow-up time to accumulate the number of UC patients. It is expected that these can increase the transition interval of MetS and MO phenotypes and analyze their impacts on the risk of development of each type of UC (kidney cancer, prostate cancer, bladder cancer) separately.

## Conclusions

The presence of MetS or MUO at baseline increased the risk of UC, and the participants with transitions of MetS to MetS, MUO to MUO and even MUO to MHO over time had a significantly higher risk of UC. These findings might provide the scientific basis for preventing UC by maintaining metabolic health and controlling obesity. In view of the fact that this study failed to analyze the impact of MetS and MO phenotypes on each type of UC, it is hoped that with the extension of follow-up time and the accumulation of case numbers, the impacts on kidney cancer, prostate cancer, and bladder cancer can be further clarified respectively.

## Supporting information

**S1 Table. Characteristics of participants by transitions in MetS status, 2006–2007 to 2008–2009.**
(DOCX)

**S2 Table. Baseline characteristics of participants by MO status, 2006–2007.**
(DOCX)

**S3 Table. Characteristics of participants by transitions in MO status, 2006–2007 to 2008–2009.**
(DOCX)

**S4 Table. Subgroup analyses of the association between MetS status at baseline (2006–2007) and risk of UC.**
(DOCX)

**S5 Table. Subgroup analyses of the association between MO status at baseline (2006–2007) and risk of UC.**
(DOCX)

**S6 Table. Subgroup analyses of the association between transitions in MetS status (2006–2007 to 2008–2009) and risk of UC.**
(DOCX)

**S7 Table. Subgroup analyses of the association between transitions in MO status (2006–2007 to 2008–2009) and risk of UC.**
(DOCX)

**S8 Table. Sensitivity analyses of the association between MetS and MO status at baseline (2006–2007) and risk of UC.**
(DOCX)

**S9 Table. Sensitivity analyses of the association between transitions in MetS and MO status (2006–2007 to 2008–2009) and risk of UC.**
(DOCX)

## Acknowledgments

We acknowledge the clinical support and help with data collection provided by The Central Laboratory of Kailuan General Hospital. A preprint has previously been published [46].

## Author Contributions

**Conceptualization:** Runxue Jiang, Shouling Wu, Hailong Hu, Haifeng Cai.

**Data curation:** Jianglun Shen, Shuohua Chen.

**Formal analysis:** Runxue Jiang, Shuohua Chen.

**Funding acquisition:** Runxue Jiang.

**Investigation:** Shuohua Chen.

**Methodology:** Runxue Jiang, Shouling Wu, Hailong Hu.

**Project administration:** Runxue Jiang, Shouling Wu.

**Resources:** Haifeng Cai.

**Software:** Shuohua Chen.

**Supervision:** Haifeng Cai.

**Validation:** Jianglun Shen.

**Visualization:** Xia Wang.

**Writing – original draft:** Xia Wang.

**Writing – review & editing:** Runxue Jiang, Jianglun Shen.

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
