## [Decision Letter · Decision Letter 0]

5 Aug 2024

PONE-D-24-26527Transitions in metabolic syndrome and metabolic obesity status over time and risk of urologic cancer: A prospective cohort studyPLOS ONE

Dear Dr. Jiang,

Thank you for submitting your manuscript to PLOS ONE. After careful consideration, we feel that it has merit but does not fully meet PLOS ONE’s publication criteria as it currently stands. Therefore, we invite you to submit a revised version of the manuscript that addresses the points raised during the review process.

We look forward to receiving your revised manuscript.

Kind regards,

Yuki Arita, M.D., Ph.D

Academic Editor

PLOS ONE

Journal Requirements:

2. In the online submission form, you indicated that The data that support the fndings of this study are available from the Kailuan Study but restrictions apply to the availability of these data, which were used under license for the current study, and so are not publicly available. Data are however available from the authors upon reasonable request and with permission of the Kailuan Study.

3. Please include your tables as part of your main manuscript and remove the individual files. Please note that supplementary tables (should remain/ be uploaded) as separate "supporting information" files

Reviewers' comments:

Reviewer's Responses to Questions

**Comments to the Author**

1. Is the manuscript technically sound, and do the data support the conclusions?

Reviewer #1: Yes

Reviewer #2: Yes

2. Has the statistical analysis been performed appropriately and rigorously? 

Reviewer #1: Yes

Reviewer #2: Yes

3. Have the authors made all data underlying the findings in their manuscript fully available?

Reviewer #1: Yes

Reviewer #2: Yes

4. Is the manuscript presented in an intelligible fashion and written in standard English?

Reviewer #1: Yes

Reviewer #2: Yes

5. Review Comments to the Author

**Reviewer #1: **Summary of the review

Study design and manuscript contents:

o This prospective cohort study investigates the association between metabolic syndrome (MetS), metabolic obesity (MO) status, their transitions over time, and the risk of developing urologic cancer (UC).

o The study is well-structured with a robust dataset of 97,897 participants and a follow-up period of approximately 14 years.

o The methodology is comprehensive, employing multifactorial Cox proportional risk regression models to assess hazard ratios.

Relevant comments, including major strengths and major weaknesses:

Strengths:

Large sample size and long follow-up period.

Detailed analysis of MetS and MO status transitions.

Comprehensive statistical adjustments for potential confounders.

Weaknesses:

Limited representation of female participants, which may affect the generalizability of results.

Lack of detailed analysis for different types of UC.

Short transition intervals for MetS and MO status may not capture long-term effects adequately.

Minor weaknesses:

o Some sections of the manuscript, particularly the introduction, could benefit from more concise language.

o The discussion section could be better organized to clearly separate findings from interpretations.

Detailed comments

Title:

1. The title is clear and informative, effectively summarizing the study's focus.

Abstract:

2. The abstract provides a concise summary of the study objectives, methods, results, and conclusions.

3. Consider clarifying the transition intervals and their significance.

Introduction:

4. The introduction provides a good overview of the background and rationale for the study.

5. It would benefit from a more focused discussion on the gaps in existing research that this study aims to address.

Materials and Methods:

6. The methods section is detailed and thorough.

7. The explanation of the multifactorial Cox proportional risk regression models is clear.

8. Consider providing more detail on the selection criteria for participants included in the transition analysis.

Results:

9. The results are presented clearly, with appropriate use of tables and figures.

10. The section on subgroup analyses is informative but could be better integrated into the overall results narrative.

Discussion:

11. The discussion effectively interprets the findings and their implications.

12. It could be improved by explicitly addressing the study's limitations and suggesting areas for future research.

13. Consider discussing potential biological mechanisms linking MetS, MO, and UC in more detail.

Conclusion:

14. The conclusion succinctly summarizes the main findings and their potential implications for UC prevention.

15. It would benefit from a brief mention of the study's limitations.

References:

16. The references are relevant and up-to-date.

17. Ensure that all references are formatted consistently according to the journal's guidelines.

Figures and Tables:

18. Figures and tables are well-designed and effectively illustrate the study's findings.

19. Consider adding more explanatory notes to the tables for clarity.

**Reviewer #2: **Thanks for allowing me to review this paper.

This study investigates the relationship between metabolic syndrome (MetS), metabolic obesity (MO), and the risk of urologic cancer (UC), offering valuable insights and clinical implications. However, several areas need improvement to fulfill the requirements of the prime journal. These include elucidating and condensing the introduction, furnishing with more specific explanations of the methods, integrating the results and discussion in a most felicitous manner, and addressing the limitations and outlining potential directions for future research more explicitly.

Specific comments are as follows:

Title:

1. The title is succinct and reflect the objectives and content clearly.

Abstract:

2. The abstract is comprehensive but would benefit from a clearer articulation about its limitations.

3. Ensure that the abstract concisely accentuate the key findings and the implications.

Introduction:

4. The introduction is well-organized but could be strengthened with a more thorough review of the literature on MetS, MO, and UC.

Methods:

5. The methods section is informative and well-structured.

6. Consider providing a brief clarification for the selected follow-up period and intervals used to assess the transitions in MetS and MO.

Results:

7. Results are demonstrated accurately based on proper statistical analysis.

8. Considering integrating the presentation of subgroup analyses more effectively and appropriately with the overall results.

Discussion:

9. The discussion puts a detailed interpretation of the results but would be enhanced with a clearer structure.

10. Consider separating the discussion of study findings from the limitations and future directions.

11. Investigating the potential mechanisms interconnecting MetS, MO, and UC would yield more valuable insights.

Conclusion:

12. The conclusion summarizes the key points in an efficient manner.

13. It could benefit from mentioning the study's limitations and the possible and potential directions for future research.

References:

14. References are appropriate and up to date.

15. Ensure consistency with journal's guidelines for reference formatting.

Figures and Tables:

16. Figures and tables are clear and well-presented.

17. Providing some additional notes with explanations of the tables could make it more understandable for readers.

6. PLOS authors have the option to publish the peer review history of their article (what does this mean?). If published, this will include your full peer review and any attached files.

Reviewer #1: No

Reviewer #2: No

---

## [Author Response · Author response to Decision Letter 0]

12 Sep 2024

Dear Editor and Reviewers:

Thank you for your careful review of our manuscript and for your valuable comments. We have carefully revised our manuscript according to your kind advices and reviewer’s suggestions. enclosed please find the responses to the referees. We sincerely hope this manuscript will be finally acceptable to be published on “PLOS ONE”. Thank you very much for all your help and looking forward to hearing from you soon.

Best regards

Sincerely yours

Dr. Runxue Jiang

Please find the following response to the comments of reviewers.

Reviewer #1: Summary of the review

Study design and manuscript contents:

o This prospective cohort study investigates the association between metabolic syndrome (MetS), metabolic obesity (MO) status, their transitions over time, and the risk of developing urologic cancer (UC).

o The study is well-structured with a robust dataset of 97,897 participants and a follow-up period of approximately 14 years.

o The methodology is comprehensive, employing multifactorial Cox proportional risk regression models to assess hazard ratios.

Relevant comments, including major strengths and major weaknesses:

Strengths:

Large sample size and long follow-up period.

Detailed analysis of MetS and MO status transitions.

Comprehensive statistical adjustments for potential confounders.

Weaknesses:

Limited representation of female participants, which may affect the generalizability of results.

Lack of detailed analysis for different types of UC.

Short transition intervals for MetS and MO status may not capture long-term effects adequately.

Minor weaknesses:

o Some sections of the manuscript, particularly the introduction, could benefit from more concise language.

o The discussion section could be better organized to clearly separate findings from interpretations.

Response: We are grateful for the effort and detailed comments from the reviewer – thank you.

Detailed comments

Title:

1. The title is clear and informative, effectively summarizing the study's focus.

Abstract:

2. The abstract provides a concise summary of the study objectives, methods, results, and conclusions.

3. Consider clarifying the transition intervals and their significance.

Response:

The interval of transition between MetS and MO phenotypes was determined based on the health examination in 2006–2007 and in 2008–2009. The period for each observation object is approximately two years. The dynamic changes of independent variables should be more reliable and convincing in their impact on dependent variables than single exposures. The main purpose of this study is to explore the impact of MetS and MO status and their dynamic changes on UC.

We modified the abstract as follows: 

Abstract: “Background & aims

The effects of metabolic obesity (MO) phenotypes status and their dynamic changes on urologic cancer (UC) is ignored. We aimed to investigate the association between metabolic syndrome (MetS) and MO status at baseline, their dynamic changes and UC risk.”

Introduction:

4. The introduction provides a good overview of the background and rationale for the study.

5. It would benefit from a more focused discussion on the gaps in existing research that this study aims to address.

Response:

We have revised the introduction part (line 76-92).

Materials and Methods:

6. The methods section is detailed and thorough.

7. The explanation of the multifactorial Cox proportional risk regression models is clear.

8. Consider providing more detail on the selection criteria for participants included in the transition analysis.

Response:

The inclusion and exclusion criteria are in the line 106-109 in the main text.

The number of participants who meet the inclusion and exclusion criteria of the cohort is 97,897.

35,178 participants, amongst whom 24,162 did not participate in the 2008−2009 survey, 34 with incident UC between the 2006−2007 and 2008−2009 surveys, 3,186 participants with missing data on BMI and 7,796 participants with missing data on MetS components in the 2008–2009 survey were further excluded, leaving 62,719 participants for assessing the association of transitions in MetS and MO status and risk of UC.

The description is provided in the line 118-123 in the main text.

Results:

9. The results are presented clearly, with appropriate use of tables and figures.

10. The section on subgroup analyses is informative but could be better integrated into the overall results narrative.

Response:

The content of this part has been integrated.

Discussion:

11. The discussion effectively interprets the findings and their implications.

12. It could be improved by explicitly addressing the study's limitations and suggesting areas for future research.

13. Consider discussing potential biological mechanisms linking MetS, MO, and UC in more detail.

Response:

“Solutions to limitations and future research directions are to establish a dynamic cohort to expand the number of observation objects and prolong the follow-up time to accumulate the number of UC patients. It is expected that this can increase the transition interval of MetS and MO phenotypes and analyze their impacts on the risk of occurrence of each type of UC (kidney cancer, prostate cancer, bladder cancer) separately ”(line 375-380).

Potential biological mechanisms have been expanded (line 332-358).

Conclusion:

14. The conclusion succinctly summarizes the main findings and their potential implications for UC prevention.

15. It would benefit from a brief mention of the study's limitations.

Response:

The following content is added to the conclusion part(line 386-390).

“In view of the fact that this study failed to analyze the impact of MetS and MO phenotypes on each type of UC, it is hoped that with the extension of follow-up time and the accumulation of case numbers, the impacts on kidney cancer, prostate cancer, and bladder cancer can be further clarified respectively.”

References:

16. The references are relevant and up-to-date.

17. Ensure that all references are formatted consistently according to the journal's guidelines.

Response:

The format of references has been adjusted according to the journal's guidelines.

Figures and Tables:

18. Figures and tables are well-designed and effectively illustrate the study's findings.

19. Consider adding more explanatory notes to the tables for clarity.

Response:

Some Abbreviations have been added. Is the table difficult to understand or in need of clarification?

Reviewer #2: Thanks for allowing me to review this paper.

This study investigates the relationship between metabolic syndrome (MetS), metabolic obesity (MO), and the risk of urologic cancer (UC), offering valuable insights and clinical implications. However, several areas need improvement to fulfill the requirements of the prime journal. These include elucidating and condensing the introduction, furnishing with more specific explanations of the methods, integrating the results and discussion in a most felicitous manner, and addressing the limitations and outlining potential directions for future research more explicitly.

Response:

Thank you for these comments.

Specific comments are as follows:

Title:

1. The title is succinct and reflect the objectives and content clearly.

Abstract:

2. The abstract is comprehensive but would benefit from a clearer articulation about its limitations.

3. Ensure that the abstract concisely accentuate the key findings and the implications.

Response:

We added the limitations of the study to the abstract.

“The main limitations of this study are as follows: the ratio of men to women in the cohort is unbalanced; the impacts of MetS and MO on each cancer type (kidney cancer, prostate cancer, bladder cancer) have not been analyzed separately; the transition intervals of MetS and MO phenotypes are relatively short.”(line 43-46)

Introduction:

4. The introduction is well-organized but could be strengthened with a more thorough review of the literature on MetS, MO, and UC.

Response:

We have revised the introduction part (line 76-91).

Methods:

5. The methods section is informative and well-structured.

6. Consider providing a brief clarification for the selected follow-up period and intervals used to assess the transitions in MetS and MO.

Response:

The association between the transitions in MetS (compared with non-MetS to non-MetS individuals) and MO status (compared with MHN to MHN individuals) from baseline (2006−2007) to 2008−2009 survey and the risk of UC incidence during the follow up (after 2008−2009) was also assessed in the same way (line 188-192).

The last follow up was completed on December 31, 2020 ( line169-170).

Results:

7. Results are demonstrated accurately based on proper statistical analysis.

8. Considering integrating the presentation of subgroup analyses more effectively and appropriately with the overall results.

Response:

The content of this part has been integrated.

Discussion:

9. The discussion puts a detailed interpretation of the results but would be enhanced with a clearer structure.

10. Consider separating the discussion of study findings from the limitations and future directions.

11. Investigating the potential mechanisms interconnecting MetS, MO, and UC would yield more valuable insights.

Response:

The structure of the discussion section has been adjusted. Future research directions have been proposed in view of the limitations of this study, and the potential mechanisms of the relationship between MetS, MO and UC have been enriched.

Conclusion:

12. The conclusion summarizes the key points in an efficient manner.

13. It could benefit from mentioning the study's limitations and the possible and potential directions for future research.

Response:

The following content is added to the conclusion part (line 386-390).

“In view of the fact that this study failed to analyze the impact of MetS and MO phenotypes on each type of UC, it is hoped that with the extension of follow-up time and the accumulation of case numbers, the impacts on kidney cancer, prostate cancer, and bladder cancer can be further clarified respectively.”

References:

14. References are appropriate and up to date.

15. Ensure consistency with journal's guidelines for reference formatting.

Response:

The format of references has been adjusted according to the journal's guidelines.

Figures and Tables:

16. Figures and tables are clear and well-presented.

17. Providing some additional notes with explanations of the tables could make it more understandable for readers.

Response:

Some Abbreviations have been added. Is the table difficult to understand or in need of clarification?

---

## [Decision Letter · Decision Letter 1]

20 Sep 2024

Transitions in metabolic syndrome and metabolic obesity status over time and risk of urologic cancer: A prospective cohort study

PONE-D-24-26527R1

Dear Dr. Jiang,

We’re pleased to inform you that your manuscript has been judged scientifically suitable for publication and will be formally accepted for publication once it meets all outstanding technical requirements.

Kind regards,

Yuki Arita, M.D., Ph.D

Academic Editor

PLOS ONE

Additional Editor Comments (optional):

All comments have been addressed and the manuscript has been much improved and is in a nice condition now.

Reviewers' comments:

Reviewer's Responses to Questions

**Comments to the Author**

1. If the authors have adequately addressed your comments raised in a previous round of review and you feel that this manuscript is now acceptable for publication, you may indicate that here to bypass the “Comments to the Author” section, enter your conflict of interest statement in the “Confidential to Editor” section, and submit your "Accept" recommendation.

Reviewer #1: All comments have been addressed

Reviewer #2: All comments have been addressed

2. Is the manuscript technically sound, and do the data support the conclusions?

Reviewer #1: Yes

Reviewer #2: Yes

3. Has the statistical analysis been performed appropriately and rigorously? 

Reviewer #1: Yes

Reviewer #2: Yes

4. Have the authors made all data underlying the findings in their manuscript fully available?

Reviewer #1: Yes

Reviewer #2: Yes

5. Is the manuscript presented in an intelligible fashion and written in standard English?

Reviewer #1: Yes

Reviewer #2: Yes

6. Review Comments to the Author

Reviewer #1: This second version of the paper is a great improvement, the authors are to be commended.

The manuscript has been much improved and is in a nice condition now.

Reviewer #2: After thorough and careful revisions, the manuscript has seen significant improvements in many aspects.

1. The abstract now briefly addresses some of the study's shortcomings, making this section more complete.

2. The introduction section has been enhanced with a brief explanation between the metabolic obesity subtypes and their potential risks of cancer, presenting the questions and laying a theoretical foundation for the subsequent sections.

3. The results section now includes additional tables, which is more informative, and integrates the results of the subgroup analysis with the overall findings, which better highlights the conclusion.

4. The discussion section has been expanded to include additional potential mechanisms, providing more comprehensive information that makes the results easier to understand.

5. The strengths and limitations section thoroughly summarizes the study's limitations and highlights future research directions that should be addressed, thereby making the manuscript more complete and the conclusion robust.

I believe the paper has now essentially met the requirements for publication, and I recommend accepting it for publication after the revisions are finalized.

7. PLOS authors have the option to publish the peer review history of their article (what does this mean?). If published, this will include your full peer review and any attached files.

Reviewer #1: No

Reviewer #2: No

---

## [Editor Report · Acceptance letter]

9 Oct 2024

PONE-D-24-26527R1 

PLOS ONE

Dear Dr. Jiang, 

I'm pleased to inform you that your manuscript has been deemed suitable for publication in PLOS ONE. Congratulations! Your manuscript is now being handed over to our production team.

Kind regards, 

on behalf of

Dr. Yuki Arita 

Academic Editor

PLOS ONE